# Ecosystem engineers drive differing microbial community composition in intertidal estuarine sediments

Adam J. Wyness[1,2]*, Irene Fortune[1], Andrew J. Blight[1], Patricia Browne[1], Morgan Hartley[1], Matthew Holden[3], David M. Paterson[1]

1 Sediment Ecology Research Group, Scottish Oceans Institute, School of Biology, University of St Andrews, East Sands, St Andrews, United Kingdom, 2 Coastal Research Group, Department of Zoology and Entomology, Rhodes University, Makhanda, South Africa, 3 Infection Group, School of Medicine, University of St Andrews, North Haugh, St Andrew, United Kingdom

* a.wyness@ru.ac.za

**Data Availability Statement:** Sequence data were deposited and are publicly available in the NCBI Sequence Read Archive (SRA) under the BioProject ID PRJNA639965.

## Abstract

Intertidal systems are complex and dynamic environments with many interacting factors influencing biochemical characteristics and microbial communities. One key factor are the actions of resident fauna, many of which are regarded as ecosystem engineers because of their bioturbation, bioirrigation and sediment stabilising activities. The purpose of this investigation was to elucidate the evolutionary implications of the ecosystem engineering process by identifying, if any, aspects that act as selection pressures upon microbial communities. A mesocosm study was performed using the well characterised intertidal ecosystem engineers *Corophium volutator*, *Hediste diversicolor*, and microphytobenthos, in addition to manual turbation of sediments to compare effects of bioturbation, bioirrigation and stabilisation. A range of sediment functions and biogeochemical gradients were measured in conjunction with 16S rRNA sequencing and diatom taxonomy, with downstream bacterial metagenome function prediction, to identify selection pressures that incited change to microbial community composition and function. Bacterial communities were predominantly Proteobacteria, with the relative abundance of Bacteroidetes, Alphaproteobacteria and Verrucomicrobia being partially displaced by Deltaproteobacteria, Acidobacteria and Chloroflexi as dissolved oxygen concentration and redox potential decreased. Bacterial community composition was driven strongly by biogeochemistry; surface communities were affected by a combination of sediment functions and overlying water turbidity, and subsurface communities by biogeochemical gradients driven by sediment reworking. Diatom communities were dominated by *Nitzschia laevis* and *Achnanthes* sp., and assemblage composition was influenced by overlying water turbidity (manual or biogenic) rather than direct infaunal influences such as grazing.

**Funding:** This work was funded by the John Templeton Foundation (https://www.templeton. org/) Grant 60501, "Putting the Extended Evolutionary Synthesis to the Test". It also received funding from the MASTS pooling initiative (The Marine Alliance for Science and Technology for Scotland, (https://www.masts.ac.uk)) and their support is gratefully acknowledged. MASTS is funded by the Scottish Funding Council (grant reference HR09011) and contributing institutions. DMP received funding from MASTS, funded by the Scottish Funding Council (grant reference HR09011), and AJW received funding under the MASTS small grant scheme (grant reference SG433). The funders had no role in study design, data collection and analysis, decision to publish, or preparation of the manuscript.

**Competing interests:** The authors have declared that no competing interests exist.

## Introduction

Intertidal zones are dynamic boundaries between land and sea with extreme environmental fluctuations in wave forces, wetting/drying, salinity, light and temperature. Yet, intertidal mudflats are biologically productive and serve multiple ecosystem services such as; carbon sequestration, nutrient cycling, provision of food, remediation of waterborne pollutants and coastal flood defence [1]. Intertidal communities are adapted to a range of substrata from soft cohesive mud, to coarse sand, gravel or rock, depending upon coastal dynamics and geological conditions. Intertidal flats are spatially homogenous compared to other marine habitats and many terrestrial habitats, however the inhabitant organisms occupy different niches with regards to sediment depth, substratum type or tidal elevation [2].

On intertidal flats macrofauna such as polychaete worms and mud shrimp are termed 'ecosystem engineers' since they alter the physical environmental by forming burrows of varying depth and complexity, casts and re-work the sediment [3, 4]. Furthermore, macrofauna ventilate their burrows by passive or active flushing, to draw overlying water into the burrow (bioirrigation). In doing so they transport solutes across the sediment water boundary and there is growing evidence that bioirrigation is a significant driver of biogeochemical cycling [5]. Intertidal ecosystem engineers are known to influence sediment properties including; redox fluctuations, oxygenation of sediments, biogeochemical cycling of nutrients, increased water content, lowering critical erosion thresholds and sediment transport [6–11].

The consequences of the ecosystem engineering processes for intertidal systems are evident in; macrofauna, macrophytes (sea grass), meiofauna (animals below 0.5 mm in size) and bacteria [12–15] with interactions often being context dependant depending on abiotic conditions such as local hydrodynamics and sediment particle size [16].

One important mutualism occurs between polychaete worms and bacteria, where extracellular exudates of the worm such as feeding nets, encourage the growth and adhesion of bacteria and algae which in turn form the diet of the macrofauna [17, 18]. In addition, bioturbation and bioirrigation activities of macrofauna have been shown to increase the distribution of bacteria with depth, alter the diversity of benthic bacteria at the sediment/water interface and stimulate metabolic activity, with increased abundance of nitrogen-cycling bacteria [10, 15].

Related to the concept of ecosystem engineering is niche construction. It is distinct from ecosystem engineering in that it deals with the evolutionary implications of the engineering process [19], rather than the ecological perspective of ecosystem engineering which is the alteration of resource availability by creating, altering or destroying habitats [20, 21].

The aim of this study was to test the hypothesis that different ecosystem engineering activities of macrofauna and microphytobenthic biofilms result in different modifications of the sediment environment through niche construction, and that these environmental modifications would exert selection pressures on diatom and bacterial communities. A response by microbial communities would be observed in the shift in relative abundance of taxa; if the selection pressures exerted by ecosystem engineering is powerful enough to affect community composition, it is implied that they will also influence natural selection at the organism level.

Three model ecosystem engineers were used. The first, *Hediste (Nereis) diversicolor* (O.F. Müller, 1776), is a polychaete that builds semi-permanent mucus lined burrow networks, and displays a wide range of feeding mechanisms such as filter and deposit feeding, scavenging and predation [22]. The second, *Corophium volutator* (Pallas, 1766), is a deposit feeding amphipod that builds networks of shallow 'U' shaped burrows [3]. Finally, by not adding macrofauna to mesocosms, the effect of an uninterrupted microphytobenthos (MPB) biofilm, was also studied. Microphytobenthos is a term for photosynthetic microbiota primarily consisting of

epipelic diatoms and cyanobacteria that are well-known for their surface sediment stabilising effects due to the exudation of extracellular polymeric substances (EPS) [23].

The first objective was to identify biogeochemical niches created by infauna in the sediment environment. It was hypothesised that the presence of bioturbating infauna would create redox and dissolved oxygen conditions different to that of the non-infauna, and thereby create selection pressures to microbial communities. The combination of both infauna was included to investigate additive or interactive effects from the two mechanisms of bioturbation.

Secondly, sediments were manually-turbated to simulate physical disturbance to determine whether infaunal influences were due to physical sediment reworking, or biogenic effects such as burrow irrigation.

## Methods

### Mesocosm setup

Sediment cores (0–10 cm depth) were collected from the mid-intertidal zone at Tayport Heath, Scotland, UK (56.440893 latitude, -2.863194 longitude) during October 2017. Sediment was sieved through 0.5 mm mesh to remove macrofauna, homogenized in brackish sea water (25 practical salinity units (PSU)) and left to settle for 24 h in darkness at 10°C before removing overlying water. The sediment collected was a mud and sand mix typical of sediments found towards the mouth of estuaries (mean particle diameter = 178 μm; median particle diameter = 182 μm; proportion <63 μm = 16%; water content = 24%; organic content = 2%). Permission was obtained to collect samples at the River Tay Site of Special Scientific Interest (SSSI) from Scottish Natural Heritage under Section 16(3) of the Nature Conservation (Scotland) Act 2004.

Acrylic mesocosms (20 cm diameter, 25 cm height) were filled with 2.5 L sediment followed by 3.8 L of brackish water (25 PSU), pouring with care onto a bubble wrap disc to ensure minimal sediment resuspension. Mesocosms were left in darkness at 10°C for 48 h with 100% of the overlying water replaced every 24 h to remove nutrients released from the sediment after sieving and homogenizing.

*Corophium volutator* and *Nereis diversicolor* were collected from the same intertidal sediment at Tayport Heath. *C. volutator* retained by 0.5 mm sieve were collected, and *H. diversicolor* between 2 and 5 cm length were picked by hand from overturned sediment. Infauna was added to four replicate mesocosms to construct the following treatments: 1- *C. volutator* (3.5 g); 2- *H. diversicolor* (3.5 g); 3- *C. volutator* (1.75 g) and H. *diversicolor* (1.75 g). Treatment 4 had no infauna added, and no physical turbation to allow MPB biofilms to develop ungrazed by infauna as with treatments 1–3, and undisturbed as with treatment 5, and is hereon referred to as the 'MPB' treatment. Treatment 5 also had no infauna added, but the surface sediment was physically disturbed (proxy for bioturbation) every 24 h. Infauna densities were similar to previous mesocosm studies [4], with 1.10 mg infauna per $cm^3$ of sediment, and 10.6 mg infauna per $cm^2$ of sediment surface area. Exact numbers of individuals were not recorded to minimise handling of infauna, however 3.5 g was equivalent to roughly 750 individuals of *C. volutator*, and 35 *H. diversicolor*, equivalent to 24,000 individuals/$m^2$ and 1110 individuals/$m^2$ respectively, which are realistic environmental abundances [24, 25]. Any dead infauna were removed when observed and replaced with individuals of similar size. Daily mortality rates of *H. diversicolor* were low with no mass mortality events. *C. volutator* were generally low (<5%), with intermittent higher mortality (first event at 26 days of ~<30%) presumably due to the expected high mortality rates during moulting events observed in *C. volutator* and other *Corophium* species in captivity [26, 27]. Water changes of 75% overlying water volume were

performed every 3 to 4 d initially, with longer intervals as the incubation progressed, with close monitoring of water chemistry quality.

Manual-turbation was performed using a hand-held rake-like apparatus consisting of an acrylic base (40 mm width, 180 mm length, 6 mm depth) fixed to the head of an M8 200 mm A2 stainless steel square neck carriage bolt using an A2 stainless steel wing nut, with a threaded plastic T-handle (S1 Fig). Eighteen stainless steel panel pins (1.5 mm x 40 mm) were inserted through two rows of 1 mm width holes drilled at a 65˚ angle from the base, leaving the pins extruding roughly 30 mm below the bottom surface of the base. Rows were 20 mm apart, with pins within each row 20 mm apart. Rows were offset by 10 mm, resulting in a pin disturbing the sediment every 10 mm at 65˚ and 115˚ to a depth of 30 mm when turned through a full circle. The manual-turbation treatment consisted of 5 complete turns every 24 hours. This process was not designed to precisely mimic the actions of infauna, rather it was to examine the effects of physical disturbance and sediment reworking without macrofaunal burrow irrigation.

Mesocosms were incubated at 12˚C with a 10:14 light:dark cycle with 14000 K Marine White FMW39TS lights (Arcadia) providing a photosynthetically active radiation (PAR) of ~ 90 $\mu$mol m$^{-2}$ s$^{-1}$ delivered to each mesocosm (water surface). To reduce any variation in the light climate, the mesocosm positions were rotated every 48 hours. Mesocosms were aerated using traditional fish tank air stones, ensuring oxygen saturation at all times. Bubbles were diverted at a 45˚ angle to create a gentle rotational current. Although these conditions do not specifically mimic intertidal conditions that the sediments were extracted from, this setup allows the comparison of the niche constructing activities of organisms capable of living inter- and subtidally, with all treatments exposed to the same conditions of good light availability and moderate salinity, similar to previous studies [28]. Mesocosms were harvested after 70 days, with overlying water turbidity measured at regular intervals (n = 11), biochemical gradient measurements at 0, 35 and 70 weeks, and samples taken for the remaining sediment characteristics, diatom and bacterial community analysis at 70 weeks.

## Sediment characteristics

Redox potential and dissolved oxygen (DO) measurements were taken using slender design, clark-type microelectrodes (Unisense, Denmark). Measurements were taken at 5 mm increments from the surface to a depth of 50 mm. Overlying water turbidity measurements were taken every week using a nephelometer (Analite 156, Novasina, Switzerland). Sediment surface stability was measured upon harvesting the mesocosms at 70 days using the cohesive strength meter [29, 30] using the 'Fine 1' program. For surface stability, cores of a 30 mm diameter were taken and fixed to the meter head so water expelled from the chamber during measurements did not interfere with other measurements being taken in the mesocosm. Results are reported as the water jet stagnation pressure (Nm$^{-2}$) at the moment of significant surface erosion. Chlorophyll a, and colloidal and total carbohydrates were analysed from contact cores taken from the surface of the sediment following the protocols outlined in the HIMOM protocols [31]. Statistical analyses were performed in SPSS v24 (IBM) and plots generated using Sigmaplot v13 (Systat Software Inc.). Data were checked for compliance with ANOVA assumptions, and post-hoc analysis was performed using Fisher's least significant difference procedure with correction for multiple comparisons.

## Diatom assemblage community analysis

Surface scrapes to a depth of 2 mm and weight of ~0.5 g were stored in 1 ml 2.5% glutaraldehyde in seawater before acid cleaning and slide preparation [32]. Briefly, diatom assemblages

were extracted and cleaned using the saturated $KMnO_4$ and sulphuric acid. After centrifugation 300 μl of washed sample was pipetted onto a slide cover slip, oven dried at 65˚C and mounted using the diatom mountant Naphrax (Brunel Microscopes, UK). Three hundred and twenty five valves were identified for each sample using the Utermöhl counting method. Species that appeared in less than 5% of samples were removed from the dataset and community comparisons were performed on Bray- Curtis similarity matrices using analysis of similarities testing (ANOSIM), similarity percentage analysis (SIMPER) and principle coordinate analysis (PCO) and alpha diversity metrics calculated within PRIMER6 [33]. Kruskall- Wallis tests with Dunn's multiple comparison post-hoc tests with corrections for multiple comparisons were performed to identify significant differences between alpha diversity metrics.

### Bacterial community analysis

To account for heterogeneity across the substratum, 8 syringe cores (diameter of 2 cm) were taken to a depth of 5 cm, capped and frozen in liquid nitrogen. Cores were ejected from the syringe whilst still frozen and sections taken with a sterile blade (0–2, 15–17, 30–32 and 45–47 mm hereafter referred to as 0, 15, 30 and 45 mm sections respectively).

DNA was extracted separately for the 8 replicate core sections using the 96 well Powersoil DNA Isolation Kit (MoBio), quantified using the Qubit HS kit (Thermofisher) and pooled. The '16S Metagenomic Sequencing Library Preparation guide, Part # 15044223 Rev. B, Illumina' was followed (see supplementary materials). The 16S rDNA v3- v4 region was amplified using PCR in triplicate and pooled before the second PCR amplification.

Raw reads were filtered, trimmed and dereplicated, paired reads merged, then denoising and chimeras removed using the DADA2 pipeline [34] within QIIME2 v2017.12 [35]. Taxa were allocated to amplicon sequence variants (ASVs) using the SILVA 128 database [36]. Sequences assigned to chloroplasts, archaea, mitochondria and reads unassigned below kingdom level were removed. Alpha-diversity metrics were calculated for treatment medians within QIIME2 at a rarefaction of 35000 sequences, where all treatment groups had reached the rarefaction curve plateau, with alpha diversity metrics analysed using the same procedure as for the diatom communities. A weighted Unifrac distance matrix [37] was exported for community comparison using analysis of similarities testing (ANOSIM) and principle coordinate analysis (PCO) within PRIMER 6 [33]. For metagenomics prediction, ASV tables were rarefied at 35105 sequences and functional abundances predicted using the PICRUSt2 pipeline [38] within QIIME2. Metagenome function was predicted using Enzyme Classification (EC) numbers. Relative abundance of EC numbers relevant to sediment function and nutrient turnover in intertidal sediments were identified through the MetaCyc database [39] and plotted using Heatmapper [40]. Sequence data were deposited and are publicly available in the NCBI Sequence Read Archive (SRA) under the BioProject ID PRJNA639965.

## Results

### Sediment function

The turbidity of overlying water was relatively constant over the incubation period. The highest turbidity was consistently observed in the manually-turbated mesocosms, followed by those containing *C. volutator* (Table 1). The mixed infauna treatment and *H. diversicolor* were more similar and those with MPB only had the lowest overlying water turbidity. Chlorophyll a content was significantly higher in MPB treatments than all others except *C. volutator*, which had an intermediate content between MPB and the remaining treatments. Colloidal carbohydrates in the MPB treatment were significantly higher than all other treatments, and total carbohydrates were similar across all treatments. Surface stability in MPB mesocoms was

**Table 1. Summary table of sediment functions.**

| Treatment | *C. volutator* | *H. diversicolor* | *C. volutator & H. diversicolor* | MPB | Manual turbation | ANOVA (df = 4,15) |
|---|---|---|---|---|---|---|
| **Turbidity Day 9 ((NTU log$_{10}$) ± SE)** | 1.40 ± 0.09 (d) | 0.58 ± 0.06 (b) | 0.84 ± 0.04 (c) | 0.36 ± 0.07 (a) | 1.91 ± 0.06 (e) | F = 91.20 $p < 0.001$ |
| **Turbidity Day 29 ((NTU log$_{10}$) ± SE)** | 1.16 ± 0.07 (c) | 0.47 ± 0.05 (a) | 0.71 ± 0.1 (b) | 0.45 ± 0.06 (a) | 1.68 ± 0.05 (d) | F = 57.12 $p < 0.001$ |
| **Turbidity Day 61 ((NTU log$_{10}$) ± SE)** | 0.53 ± 0.05 (b) | 0.33 ± 0.05 (ab) | 0.32 ± 0.04 (a) | 0.27 ± 0.04 (a) | 1.51 ± 0.12 (c) | F = 59.38 $p < 0.001$ |
| **Chlorophyll a (μg g$^{-1}$ ± SE)** | 20.07 ± 2.07 (ab) | 13.87 ± 1.15 (a) | 15 ± 1.07 (a) | 26.09 ± 2.04 (b) | 15.74 ± 4.34 (a) | F = 4.23 $p < 0.017$ |
| **Colloidal carbohydrates (μg g$^{-1}$ ± SE)** | 175.3 ± 25.31 (a) | 78.60 ± 6.83 (a) | 80.90 ± 14.00 (a) | 503.99 ± 124.79 (b) | 171.43 ± 107.36 (a) | F = 5.48 $p < 0.006$ |
| **Total carbohydrates (mg g$^{-1}$ ± SE)** | 12.8 ± 0.84 | 11.11 ± 1.99 | 9.17 ± 0.54 | 15.61 ± 1.56 | 12.50 ± 2.69 | F = 1.92 $p < 0.160$ |
| **Stability (N m$^{-2}$ ± SE)** | 0.52 ± 0.11 | 0.50 ± 0.10 | 0.52 ± 0.14 | 3.26 ± 2.48 | 0.68 ± 0.16 | F = 1.18 $p < 0.358$ |

Means (n = 4) ± SE and ANOVA summary. Where NTU = nephelometric turbidity units; N m$^{-2}$ = Newton/square meter. Carbohydrates expressed as μg of glucose and standard equivalent per g of dry weight sediment. Letters below means denotes significant groups at $p < 0.05$.

higher than all others however this was not significant due to high variation in measurements (Table 1).

## Biochemical gradients

The depth profiles of oxygen was similar across all treatments, with oxygen present to a depth of 25–30 mm, a higher concentration nearer the surface, and negligible or zero levels below a 40 mm depth (some negative concentrations were recorded after data manipulation due to high sensor variability at very low concentrations). In the upper 20 mm, dissolved oxygen (DO) was highest when both infauna were present, followed by mesocosms containing only *C. volutator*. Negligible concentrations of DO were measured in mesocosms containing only *H. diversicolor* or no infauna below a depth of 10mm, except when burrow lumen were frequently encountered (e.g. high variability in infaunal treatments and unusually high means at 25 mm for *H. diversicolor* and 15 mm for *C. volutator* (Fig 1A and 1B respectively)). Dissolved oxygen in the sediments between 0 and 30 mm was considerably lower after 70 days (Fig 1B) than after 35 days (Fig 1A).

Redox potentials were higher near the sediment surface, becoming lower and treatments more similar to one another as depth increased (Fig 1C and 1D). After 35 days the highest redox potentials at each depth were observed in manually-turbated mesocosms, and those containing mixed infauna, followed by those containing the infauna individually, then MPB only mesocosms. After 70 days, *C. volutator* mesocosms, manually-turbated and mixed infauna were the highest, followed by *H. diversicolor*, then MPB only. Similar to DO concentrations, redox potentials were much lower after 70 days (Fig 1D) than those observed after 35 days (Fig 1C).

## Diatom diversity

There were no significant differences between the alpha diversity metrics of the diatom communties, with the exception of *C. volutator* and manual turbation mesocosms displaying a higher Pielou's evenness of species than *H. diversicolor* and MPB mesocosms (Dunn's post-hoc tests: z = 2.27, $p = 0.012$; z = 2.21, $p = 0.014$ for *C. volutator* vs *H. diversicolor* and MPB respectively. z = 2.31, $p = 0.010$; z = 2.7, $p = 0.012$ for manual turbation vs *H. diversicolor* and

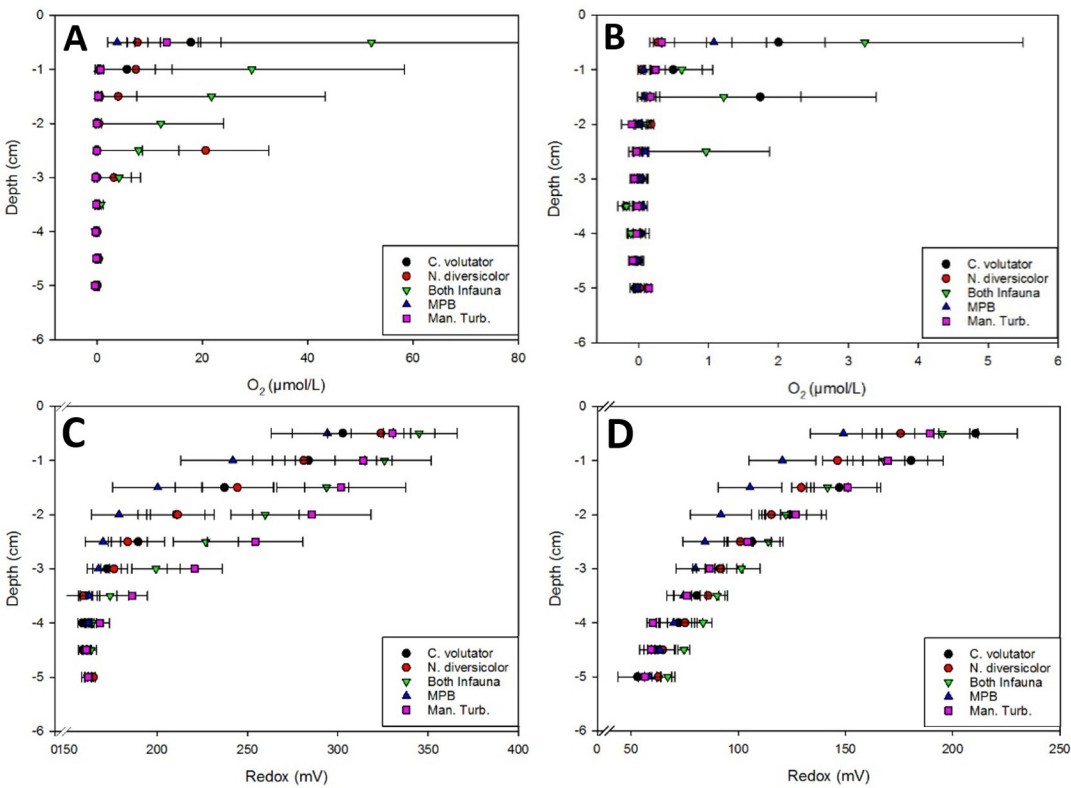

**Fig 1. Dissolved oxygen (A,B) and redox potential (C,D) profiles of mesoscosm sediment after 35 days (A,C) and 70 days (B,D) incubation.** Error bars represent 1 standard deviation from the mean (n = 6).

MPB respectively). Microphytobenthos-only mesocosms contained the highest richness of diatom species, but also relatively low diversity and evenness compared to other treatments (Table 2) due to a domination of *Nitzschia laevis* with a relative abundance of 32.1% (S1 and S2 Tables). Manually-turbated and *C. volutator* sediments had the highest diversity and evenness of diatom assemblages. There were higher species richness, and a higher diversity and evenness in mesocosms containing *C. volutator* than for *H. diversicolor* which had the lowest richness and diversity of all treatments. The mixed treatment had a higher species richness than either single infauna treatment, and diversity and evenness was intermediate between each single infauna treatment.

**Table 2. Alpha diversity metrics for diatom assemblages at the sediment surface.**

|  | Total individuals | Total species | Shannon's diversity (log e) | Pielou's evenness |
|---|---|---|---|---|
| *C. volutator* | 313.5 | 33.5 | 3.05 | 0.87 |
| *H. diversicolor* | 314.0 | 30.5 | 2.60 | 0.78 |
| Mixed | 311.5 | 33.5 | 2.83 | 0.81 |
| MPB | 308.5 | 34.0 | 2.73 | 0.78 |
| Man. Turb. | 303.0 | 32.5 | 2.98 | 0.86 |
| Kruskall- Wallis test | H (4,15) = 7.54 | H (4,15) = 2.30 | H (4,15) = 6.50 | H (4,15) = 10.64 |
|  | $p = 0.110$ | $p = 0.682$ | $p = 0.165$ | $p = 0.031$ |

Metrics summarised using treatment medians. Mixed–Mixed infauna; MPB–Microphytobenthos only; Man. Turb.–Manual-turbation.

## Diatom Diversity

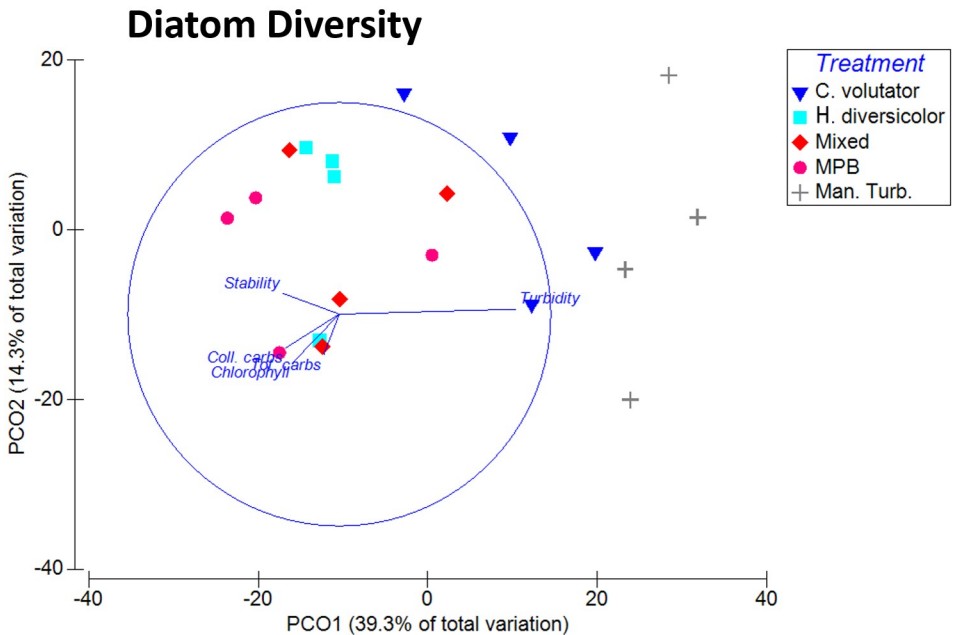

**Fig 2. Principal coordinate analysis of the beta diversity of diatom assemblages of mesocosm sediments at the sediment surface.** Ordination after standardisation of species counts and a Bray-Curtis similarity matrix. Sediment characteristics are overlaid as vectors. *C. v.–C. volutator; H. d.–H. diversicolor*; Mixed–Mixed infauna; MPB–Microphytobenthos only; Man. Turb.–Manual-turbation.

Differences in community composition were driven by a small number of diatom species (S2 Table). Differential abundance of *Nitzschia laevis* contributed the most to the pairwise dissimilarity between manual turbation and all other treatments, and between the two single infauna treatments, the treatments that were identified as significantly different (ANOSIM; S3 Table). *N. laevis* had a much lower abundance in the manually-turbated treatments than all others, and a slightly lower abundance in the *C. volutator* mesocosms. There was also a larger abundance of *Achnanthes lanceolata* in manually-turbated sediments than all others (S1 and S2 Tables).

In terms of community analysis, the assemblages inhabiting manually-turbated sediments were significantly different from all other treatments (ANOSIM; R statistics 0.43–1.00, $p = 0.029$), and the single infauna treatments were significantly different from one another (S3 Table). There was slight overlap between community compositions of MPB mesocosms and the treatments with infauna present, although they were more similar to *H. diversicolor* and the mixed treatment than *C. volutator* (Fig 2). Communities of the mixed treatment were more similar to *H. diversicolor* than *C. volutator*. PCO1, which explained 39.3% of the total variation, was highly correlated with increasing overlying water turbidity (Fig 2).

### Bacterial diversity

The largest variation between treatments for amplicon sequence variant richness (ASVs), Shannon's diversity and Faith's phylogenetic diversity were at the sediment surface (S4 Table). Although differences between depth categories were not significantly different (Kruskall-Wallis; H (3,74) = 7.45, $p = 0.059$), communities at the surface generally had lower richness than subsurface sediments (ASVs, surface median- 1668, subsurface median- 1900). Surface sediments were, however, significantly less diverse and less even than sediments at other depths

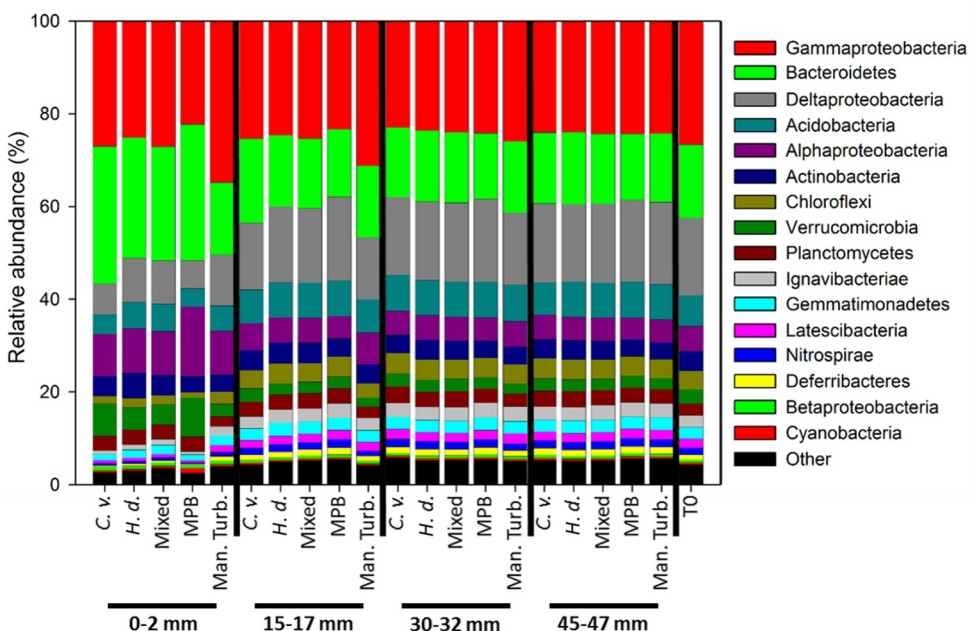

**Fig 3. Bacterial community composition for mesocosm sediments at 0, 15, 30 and 45 mm depths.** Taxa are at phylum level except Proteobacteria which are at class level. The community composition of the source sediment is included as T0. *C. v.*–*C. volutator; H. d.*–*H. diversicolor*; Mixed–Mixed infauna; MPB–Microphytobenthos only; Man. Turb.–Manual-turbation.

(Kruskall-Wallis; Shannon's diversity: H (3,74) = 29.19, $p < 0.001$. Faith's phylogenetic diversity: H (3,74) = 8.35, $p = 0.047$. Pielou's evenness: H (3,74) = 47.60, $p < 0.001$, (S4 Table)).

Bacterial community composition at phylum level was very similar between treatments, with a domination of proteobacteria at the class level (Fig 3). Surface communities had a larger relative abundance of Bacteroidetes, Alphaproteobacteria and Verrucomicrobia, and less Deltaproteoacteria and Chloroflexi than sediments below the surface. Manually-turbated sediments had a larger proportion of Gammaproteobacteria than all other sediments, regardless of depth.

The weighted Unifrac distance between community compositions of samples ranged between 0.03 and 0.30. ANOSIM analysis revealed significant differences between treatments across all depths (ANOSIM; S5 Table), and between depths across all treatments (ANOSIM; S6 Table). All pairwise comparisons between treatment groups were significant with the exception of *H. diversicolor* and Mixed (ANOSIM; S5 Table). Communities across all treatments at different depths were all significantly different from one another, except 30 and 45 mm (ANOSIM; S6 Table). The PCO clearly demonstrated that bacterial communities became more similar within and across treatments with increasing depth (Fig 4A). PCO1 explains 69.9% of the total variation, and correlates strongly with a decrease in redox potential and DO concentration representing greater sediment depth of the samples (Fig 4A). Manually-turbated sediments were separated on PCO2 (9.6% of total variation), with manually-turbated sediments at a greater depth again more similar to the other treatments.

Surface communities were significantly different between treatments (ANOSIM; S7 Table). Individual pairwise comparisons of surface sediments were all significantly different with strong group separation (S7 Table, Fig 4B), with the only exception again between *H. diversicolor* and Mixed. Overlying water turbidity correlated highly with PCO1 (63.0% of total variation), along which manually-turbated samples were separated. All vectors contributed

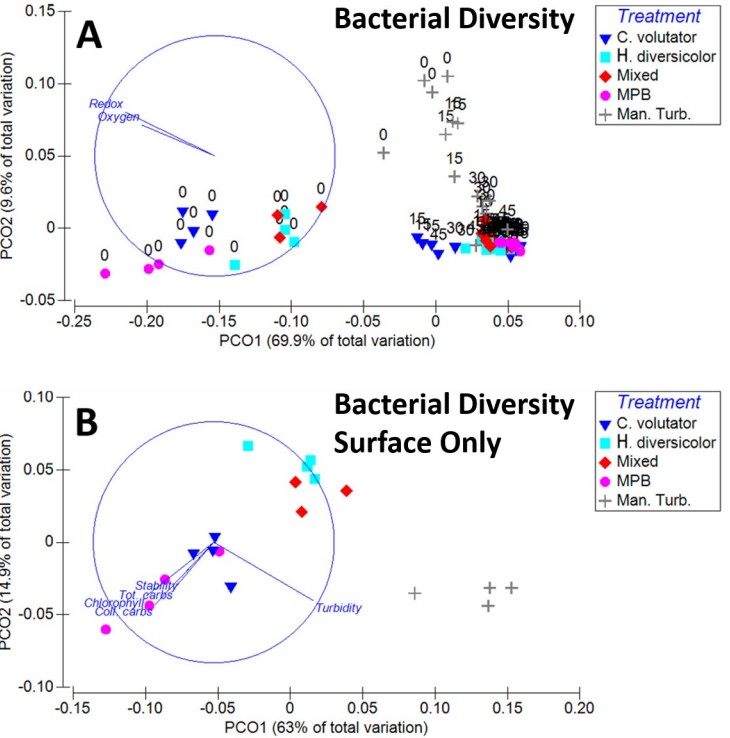

**Fig 4. Principal coordinate analysis of the beta diversity of bacterial community assemblages of all treatments at all depths (A) and the surface only (B) with overlaid vectors of biochemical gradients (A) and sediment function (B).** Ordination after standardisation of ASVs and a weighted Unifrac distance matrix. Redox potential and dissolved oxygen are overlaid as vectors for all sediment samples (A), and sediment characteristics are overlaid as vectors for sediment surface samples (B). *C. v.–C. volutator; H. d.–H. diversicolor*; Mixed–Mixed infauna; MPB–Microphytobenthos only; Man. Turb.–Manual-turbation.

similarly to PCO2 (14.9% of total variation) with infaunal and MPB treatments being separated along the common vector of colloidal and total carbohydrates and chlorophyll content, and stability.

Differences between communities between treatments at 15, 30 and 45 mm depths, and between depth groups (excluding 0 mm) were significant (ANOSIM; S8 and S9 Tables respectively). All treatment groups below the surface were significantly different with the exception of *H. diversicolor* and Mixed.

Communities at 15 mm across all treatments were different from both 3.0 and 4.5 mm (ANOSIM; S8 Table) when sub-surface sediments were plotted separately (S2 Fig), manually-turbated sediments were separated along PCO1 (31.6% of total variation), which correlated more strongly with redox than DO concentration. *C. volutator* sediments were separated along PCO2 (14.0% of total variation), which correlated more strongly with DO concentration than redox potential.

## Predicted bacterial metagenome function

Metagenomic function was profiled using gene family predictions from taxa allocated to the 16S rRNA markers of ASVs, and the nearest sequenced genome from a reference database using PICRUSt2 [38]. The core output of this analysis is an abundance table of Enzyme Classification (EC) numbers, and their higher level pathways, both of which can then be investigated

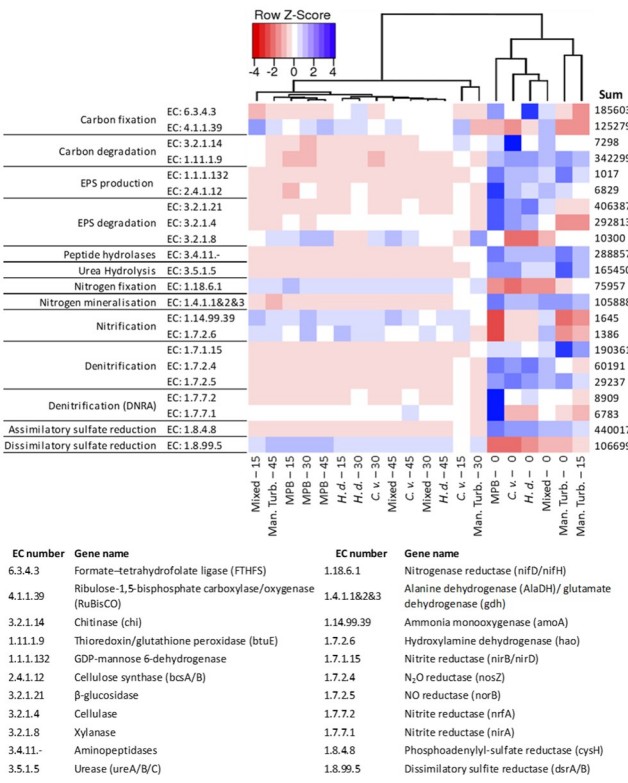

**Fig 5. Relative abundance heatmap of metagenome function as predicted by PICRUSt2.** Shading indicates z score of predicted gene family abundance for Enzyme Classification (EC) numbers (rows) between samples (columns). Samples clustered using average linkage and euclidean distances between samples. Ortholog function and EC numbers are displayed on the left of the heatmap, and sum of predicted orthologs across all samples are displayed on the right. EC numbers and gene names are displayed below. Where *C. v.–C. volutator; H. d.–H. diversicolor*; Mixed–Mixed infauna; MPB–Microphytobenthos only; Man. Turb.–Manual turbation, and the numbers preceding indicate the depth of which samples were taken (mm).

using the MetaCyc database [41]. The EC numbers featured in Fig 5 were selected for in-depth analysis because of their relevance to biogeochemical cycling in intertidal sediments.

The metabolic functions of all surface communities and manual-turbation at 15 mm were clustered together, separated from subsurface communities, reflecting patterns observed in the PCO (Fig 4A). Furthermore, sediments not exposed to the surface or turbation (manual turbation at 45 mm, below the reach of the rake, and MPB below the surface) were all more similar to each other than other samples. The extremities of abundance were almost exclusively observed in surface sediment communities (Fig 5). For the EC numbers selected, a higher presence of functional capacity was observed in surface sediments compared to subsurface sediments, with the exception of nitrification and dissimilatory sulfate reduction.

The functional capacity to produce EPS was highest in surface communities (Fig 5), as were EPS degradation genes for β- glucosidase and cellulose (EC 3.2.1.21 and 3.2.1.4 respectively) in infaunal and MPB surface sediments, but not manually-turbated sediments. The two EC numbers involved in carbon fixation showed opposite trends to one another, with RuBisCo genes (EC: 4.1.1.39) in lower abundance at the surface than subsurface sediments, and formate–tetra-hydrofolate ligase genes (EC: 6.3.4.3) highest in *H. diversicolor*, MPB and mixed sediments at the surface. Carbon degradation, peptide and urea hydrolases, nitrogen mineralisation and denitrification, and assimilatory sulfate reduction genes were generally higher in surface sediments and with manual-turbation at the surface and 15 mm. Nitrogen fixation, nitrification

and dissimilatory sulfate reduction genes were generally lower in surface sediments. Dissimilatory nitrate reduction to ammonium (DNRA) denitrification genes were in very high abundance in MPB surface sediments.

## Discussion

This study presents evidence for the alteration of biochemical gradients and sediment function by ecosystem engineers that not only effect diatom and bacterial assemblage diversity and composition, but alter the functional capacity of bacterial populations in intertidal sediments. Whilst this is not empirical proof of niche construction affecting genomic change, it is expected to occur if community change due to biochemical selection pressure arises.

Distinct niches were created by the infauna treatments compared to the MPB only mesocosms. The presence of *C. volutator* increased dissolved oxygen and redox potential in near-surface sediments, and increased sediment turnover resulting in the higher overlying water turbidity. The presence of *H. diversicolor* resulted in increased oxygen and redox potentials in sediments similarly to *C. volutator*, however microorganisms were not exposed to the same sediment reworking and suspension. The combination of both infauna created a further biogeochemical niche of higher dissolved oxygen levels and higher redox potentials than any single infauna treatment, whilst also exposing microorganisms to moderate levels of bioturbation.

We provide further evidence of conflicting ecosystem engineering effects between macrofauna and primary producers [16]. In this case, MPB are the stabilising force, through the exudation of EPS (approx 500 µg g$^{-1}$ glucose equivalents), enhancing sediment stabilisation. The presence of macrofauna was accompanied by lower EPS concentrations and lower sediment stability, and a relative increase in redox potential and DO in subsurface sediments; changes likely due to bioturbatory activity. The ratio of these different classes of ecosystem engineers (stabilisers vs turbators) in intertidal sediments represents an important dynamic that is likely to vary seasonally [42] with growth conditions for MBP and with recruitment and predation of macrofauna, adding to the variable nature of intertidal sediment behaviour.

The manual turbation of sediments using the previously described apparatus (S1 Fig) was successful in recreating the sediment reworking aspects of bioturbation without the bioirrigation of sediments. Turbidity was comparable to the *C. volutator* treatment, with very similar values for the majority of the experiment. In the final stages of the experiment turbidity in the manually turbated mesocosms were higher, presumably due to the decrease in sediment reworking by *C. volutator* as burrows had become established and were stable. Dissolved oxygen concentrations in the manually turbated sediments below the surface were negligible, whereas treatments containing infauna had higher concentrations due to burrow irrigation. The redox potential of sediments were similar between the manually turbated sediment and those containing infauna, demonstrating that redox was driven primarily by sediment reworking.

Manual-turbation and *C. volutator* increased the redox potential of sediments, however manual turbation did not increase DO concentrations. Both single infauna treatments and the mixed infauna treatment increased DO concentrations above MPB and manually-turbated sediments however, suggesting redox was governed by sediment turnover, and DO by diffusion through burrow irrigation. The extent of sediment turnover was reflected in the overlying water turbidity measurements. It affected redox potentials rather than DO, which in turn affected bacterial community composition. It was clear that manually-turbated sediments at 15 and 30 mm, and *C. volutator* sediments at 15 mm are less similar to the bulk of the subsurface sediments (Fig 4A). This compares well to the redox data, where manually turbated sediments have an increased redox potential up to a depth of 35 mm, just below the depth of which the

pins reached. This is especially evident at the mid-point of the experiment (Fig 1C). In addition, *C. volutator* also has one of the highest redox potentials at each point up to 20 mm at the end of the incubation

Also, manual turbation was found to have similar effects to *C. volutator* on the turbidity of overlying water, and EPS and chlorophyll content of the sediment. This is not surprising given the burrow cleaning activity of *C. volutator* leading to active sediment resuspension. The correlation of turbidity with the largest principle components driving both diatom and bacterial community change between treatments (Figs 2 and 4B) suggests that the physical sediment reworking (that results in increased turbidity) was a key factor in determining both diatom and bacterial community composition at the sediment surface. Sediment reworking increases light attenuation in the water column and increases nutrient release to the sediment surface and overlying water. Light attenuation was the major driver in diatom assemblage change; more so than direct biogenic influences such as selective grazing [6]. Both light and nutrient concentrations have been demonstrated to drive community change in freshwater [43], and estuarine systems [44]. However, bacterial community composition at the surface did not just correlate with turbidity as observed in diatom assemblages. Rather, the other biogenic influences of sediment stability, chlorophyll and carbohydrate concentration correlated well with community shifts, suggesting biogenic influence in addition to physical reworking. This influence may result from infaunal grazing behaviour or an interaction with diatom assemblages, in addition to being at the sediment- water interface, where effects of increased nutrient flux from burrow irrigation would be more apparent [39, 45] as well as the increased biogenic influence. Increased sediment stability, chlorophyll and carbohydrate content are strong indicators of an increased MPB biofilm content [46] therefore the trends observed in the bacterial communities of the surface sediment suggest that bacterial communities change in response to the concentration of MPB biofilm presence, regardless (at least in this case) of the MPB community dynamics.

The response of microbial species composition to the combined macrofauna treatment was not a simple additive effect of *H. diversicolor* plus *C. volutator*, but rather was skewed in favour of *H. diversicolor* treatments. This may be due to the dominance of species-specific bioirrigation potential of the polychaete over the amphipod (i.e. body mass, broad feeding style, burrow size and depth [5]. The exclusion of one or other of the macrofauna spp. reduced bacterial diversity, compared to the mixed treatment where both macrofauna were present. This supports the evidence linking biodiversity and ecosystem function and indicates, in this case, that the presence of macrofauna favours microbial diversity despite bacteria forming part of the macrofauna diet. Greater bacterial diversity may be due to the production of EPS by macrofauna (in the manner of microbial 'gardening') or reduced competition among bacteria for organic substrates or niche space. As observed in this study, loss or change of ecosystem engineers can alter microbial function and nutrient cycling in relatively short time periods. This has implications for coastal restoration projects since alterations in nutrient cycling may encourage coastal eutrophication or decrease primary production with further consequences for the food web and carbon storage [47].

It is evident in the data that there are two clusters representing the similarity of the diatom communities between each treatment. There are similarities between the *C. volutator* treatment and the manually-turbated treatment, and then similarities between the *H. diversicolor*, mixed and MPB treatments. The first of these groups had the highest diversity. Environmental conditions such as light, nutrient availability and bioturbation have a strong effect on diatom communities [48]. The different bioturbation mechanisms are causing variations in the community compositions of estuarine diatoms in this study, with turbidity. The presence of different bioturbators is altering the environmental conditions which is subsequently causing these

observed shifts in the community. As mentioned, the manually-turbated treatment and *C. volutator* treatment have the highest similarity between diatom communities. These treatments have a notably higher turbidity which factors into the difference due to the reduced light availability. As photosynthetic organisms, light availability is directly linked to the success of diatom communities.

*Nitzschia laevis* was the most influential species in separating treatment groups. It is a tolerant, motile diatom species which can survive harsh conditions such as high levels of pollution and it has even been shown to grow in the complete absence of light [49]. Therefore, it could be assumed that there would be a high survival of this species across all treatments. However, in this study it appears that this species struggled to cope with the high turbidity, hence the reduction in its presence in both the manual turbation and *C. volutator* treatments. *Nitzschia laevis* appears to be a strong competitor in the treatments with lower turbidity (MPB, *H. diversicolor*, and mixed), shown by its high relative abundance. The reduction of this species in the turbid treatments is likely linked to the higher diversity found here. The lack of this dominant species allows different species to thrive.

The chlorophyll a content of sediments can be used to represent the biomass of microphytobenthos in each treatment. The most successful diatom population in term of biomass was the MPB treatment, seemingly due to the lack of disturbance occurring in this mesocosm. The remaining treatments contained a similar amount of chlorophyll with the exception of *C. volutator* which was between the MPB and remaining treatments, suggesting the combination of bioturbation and bioirrigation was beneficial to diatom populations through the increased cycling of nutrients, specifically *Nitzchia closterium* which thrived in this treatment.

Bacterial diversity was examined at different sediment depths, since the influence of ecosystem engineers (bioturbation and bioirrigation) is variable with depth and macrofaunal species identity, as demonstrated in the redox profiles. The bacterial community composition across all mesocosms was similar to those previously reported through the use of NGS techniques on intertidal sediments collected at a similar depth from a geographically close estuary in Scotland [47], and intertidal mudflat samples from Marennes-Oléron Bay in western France [50]. Where, similarly to the results presented here, there was a domination by Proteobacteria (predominantly Gammaproteobacteria, Deltaproteobacteria and Alphaproteobacteria), and Bacteroidetes.

Bacterial community composition observed in this study was more variable at the sediment surface and driven predominantly by overlying water turbidity and proxies for biofilm production. Communities were more similar with an increase in sediment depth, correlating with a decrease in redox potential and dissolved oxygen concentrations. Community metagenome function also followed this trend, with anaerobic nutrient-cycling pathways more abundant in sub-surface and non-turbated sediments. Bacterial diversity was lower in surface sediments, leading to the extremes of the relative abundance of most of the investigated metagenome function occurring at the sediment surface.

Burrow irrigation is well-documented to increase nitrification-denitrification cycles, increase flux of ammonium to the surface and overlying water, and enhance remobilization of many other soluble elements [5, 51, 52]. There was no overriding evidence of this in the gene function analysis, perhaps due to the sampling method (not sampling in close proximity to the burrow wall), or the incubation period was not long enough for selection pressures to stabilise at depth in the sediment. However, bacterial communities in manually-turbated sediments at 15 mm had similar abundances of nitrogen cycling genes to surface sediments, suggesting that sediment reworking rather than biogenic effects may be an important factor. However, the abundance of the amoA gene (EC: 1.14.99.39), the first stage in nitrification (ammonium to hydroxylamine) was lower in surface sediments treatments without infauna.

Redox potentials were lower in sediments where there was no turbation. This included subsurface MPB sediments, and also manual turbation at 45 mm. Dissimilatory sulfate reduction (EC: 1.8.99.5) was higher in these sediments, correlating with an increase in the Delataproteobacteria, which contains many anaerobic sulfate and sulfur reducing bacteria. The absence of bioturbation allowed a stable anoxic zone, and the development of more negative redox conditions which allows greater relative abundance of these bacteria. The functional capacity for dissimilatory nitrate reduction to ammonium (DNRA) was much higher in MPB surface sediments than any other treatments. The importance of this pathway in estuarine nitrogen cycles has recently become more realised and the occurrence of DNRA bacteria have been demonstrated to be highly heterogeneous [53]. Therefore, such variability in DNRA bacteria in response to grazing pressure in this study is of great interest. DNRA is an anaerobic process, and the relatively low oxygen in the MPB treatments due to lack of bioirrigation may have resulted in an increase in the relative abundance of bacteria with DNRA capacity.

There was a higher functional capacity observed for many EC numbers in the surface sediments compared to the subsurface. This is perhaps surprising considering there was a lower ASV richness and diversity on the sediment surface than the subsurface, and with lower richness and diversity, lower functional capacity would be expected.

Acetogenesis (EC: 6.3.4.3, FTHFS) was markedly higher in *H. diversicolor* sediments than other treatments at every depth. Acetogenesis is an important stage in the carbon cycle as it greatly exceeds production of methane and is estimated to produce $10^{13}$ kg of acetate (for further metabolism) annually in anaerobic environments globally [54]. The striking increase in acetogenesis in *H. diversicolor* and MPB treatments, on the sediment surface, did not seem to correlate with any sediment functions measured in this study, therefore it may be associated with specific exudations of *H. diversicolor*, such as their mucus feeding webs that extend through their galleries and across the sediment surface. MPB EPS, which is produced in order to stabilize sediments, may have influenced this in a similar manner.

Bacteria with the potential to utilise chitin (EC: 3.2.1.14) were more abundant in *C. volutator* treatments, presumably capitalising on moulted exoskeletons of the amphipod. There was a lower abundance of the gene in the mixed treatment, which suggests a potential competition for chitin as a 'food' source between the scavenging of *H. diversicolor* for moulted exoskeletons and for bacteria. EPS production and degradation were both at their highest at the sediment surface. Cyanobacteria are a significant proportion of MPB biofilms, and contribute to the EPS matrix. The abundance of bacteria able to degrade EPS highlights the importance of EPS as a carbon source for microbial systems within sediments, and also the microbial role in sediment carbon cycling [55].

In conclusion, this study demonstrates conflicting ecosystem engineering outcomes between macrofauna and primary producers, in this case MPB. Furthermore, we provide strong evidence of rapid change in intertidal microbial community composition and function in response to ecosystem engineering by benthic macrofauna. This reveals the vulnerability of microbial metabolism, as a proxy for key ecosystem functions and services, to changes in macrofaunal assemblages and demonstrates their importance of ecosystem engineers to sustaining functional coastal systems.

## Supporting information

**S1 Fig. Apparatus used for the manual turbation of sediments.** It consists of an acrylic base (40 mm width, 180 mm length, 6 mm depth) fixed to the head of an M8 200 mm A2 stainless steel square neck carriage bolt using an A2 stainless steel wing nut, with a threaded plastic T-

handle. Eighteen stainless steel panel pins (1.5 mm x 40 mm) are inserted through two rows of 1 mm width holes drilled at a 65˚ angle from the base, leaving the pins extruding roughly 30 mm below the bottom surface of the base. Rows were 20 mm apart, with pins within each row 20 mm apart. Rows were offset by 10 mm, resulting in a pin disturbing the sediment every 10 mm at 65˚ and 115˚ to a depth of 30 mm when turned through a full circle.
(TIF)

**S2 Fig. Principle coordinate analysis of the beta diversity of bacterial community assemblages of all treatments for subsurface sediments at depths of 15, 30 and 45 mm.** Ordination after standardisation of ASVs and a weighted Unifrac distance matrix. Redox potential and dissolved oxygen are overlaid as vectors. *C. v.–C. volutator; H. d.–H. diversicolor*; Mixed–Mixed infauna; MPB–Microphytobenthos only; Man. Turb.–Manual-turbation.
(TIF)

**S1 Table. Relative abundance of diatom species identified in surface sediments.**
(DOCX)

**S2 Table. SIMPER analysis of pairwise treatments identified as significantly different by ANOSIM.** Species included up to a cumulative contribution to dissimilarity of 40%. *C. v.–C. volutator; H. d.–H. diversicolor*; Mixed–Mixed infauna; MPB–Microphytobenthos only; Man. Turb.–Manual-turbation. Rel. abun.–relative abundance; mean abun. diff.- mean abundance difference; diss.- dissimilarity.
(DOCX)

**S3 Table. ANOSIM summary table for diatom assemblage composition between treatment groups.** Sample statistic (global R): 0.47, *p* = 0.001. *C. v.–C. volutator; H. d.–H. diversicolor*; Mixed- Mixed infauna; MPB- Microphytobenthos only; Man. Turb.- Manual turbation.
(DOCX)

**S4 Table. Alpha diversity metrics for bacterial community assemblages.** Sequences rarefied at 35000 reads and metrics summarised using treatment medians. *C. v.–C. volutator; H. d.–H. diversicolor*; Mixed–Mixed infauna; MPB–Microphytobenthos only; Man. Turb.–Manual-turbation. Metrics for source sediment included as T0.
(DOCX)

**S5 Table. ANOSIM summary table for bacterial assemblage composition between all treatment groups across all depths.** Sample statistic (global R): 0.51, *p* = 0.001. *C. v.–C. volutator; H. d.–H. diversicolor*; Mixed- Mixed infauna; MPB- Microphytobenthos only; Man. Turb.- Manual turbation.
(DOCX)

**S6 Table. ANOSIM summary table for bacterial assemblage composition between all depths across all treatment groups.** Sample statistic (global R): 0.53, *p* = 0.001. *C. v.–C. volutator; H. d.–H. diversicolor*; Mixed- Mixed infauna; MPB- Microphytobenthos only; Man. Turb.- Manual turbation.
(DOCX)

**S7 Table. ANOSIM summary table for bacterial assemblage composition between treatment groups for surface sediments only.** Sample statistic (global R): 0.90, *p* = 0.001. *C. v.–C. volutator; H. d.–H. diversicolor*; Mixed- Mixed infauna; MPB- Microphytobenthos only; Man. Turb.- Manual turbation.
(DOCX)

**S8 Table. ANOSIM summary table for bacterial assemblage composition between all treatment groups across all depths for subsurface sediments only.** Sample statistic (global R): 0.28, $p$ = 0.001. *C. v.–C. volutator*; *H. d.–H. diversicolor*; Mixed- Mixed infauna; MPB- Microphytobenthos only; Man. Turb.- Manual turbation.
(DOCX)

**S9 Table. ANOSIM summary table for bacterial assemblage composition between all depths across all treatment groups for subsurface sediments only.** Sample statistic (global R): 0.39, $p$ = 0.001. *C. v.–C. volutator*; *H. d.–H. diversicolor*; Mixed- Mixed infauna; MPB- Microphytobenthos only; Man. Turb.- Manual turbation.
(DOCX)

# Acknowledgments

The authors would like to thank Dr Andrew Spiers at the University of Abertay for assistance and loan of the microelectrode apparatus.

# Author Contributions

**Conceptualization:** Adam J. Wyness, Andrew J. Blight, Matthew Holden, David M. Paterson.

**Data curation:** Adam J. Wyness, Patricia Browne, Morgan Hartley.

**Formal analysis:** Adam J. Wyness, Patricia Browne, Morgan Hartley, David M. Paterson.

**Funding acquisition:** Adam J. Wyness, Andrew J. Blight, Matthew Holden, David M. Paterson.

**Investigation:** Adam J. Wyness, Andrew J. Blight, Patricia Browne, Morgan Hartley, Matthew Holden, David M. Paterson.

**Methodology:** Adam J. Wyness, Andrew J. Blight, Morgan Hartley, Matthew Holden, David M. Paterson.

**Project administration:** Adam J. Wyness, Andrew J. Blight, Matthew Holden, David M. Paterson.

**Resources:** Adam J. Wyness, Andrew J. Blight, Matthew Holden, David M. Paterson.

**Software:** Andrew J. Blight, Matthew Holden, David M. Paterson.

**Supervision:** Andrew J. Blight, Matthew Holden, David M. Paterson.

**Validation:** Adam J. Wyness, Andrew J. Blight, Matthew Holden, David M. Paterson.

**Visualization:** Adam J. Wyness, Matthew Holden.

**Writing – original draft:** Adam J. Wyness, Irene Fortune, Andrew J. Blight, Morgan Hartley, Matthew Holden, David M. Paterson.

**Writing – review & editing:** Adam J. Wyness, Irene Fortune, Andrew J. Blight, Morgan Hartley, Matthew Holden, David M. Paterson.

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
