## [Decision Letter · Decision Letter 0]

17 Nov 2020

PONE-D-20-31006

Ecosystem engineers drive differing microbial community composition in intertidal estuarine sediment

PLOS ONE

Dear Dr. Wyness,

Thank you for submitting your manuscript to PLOS ONE. After careful consideration, we feel that it has merit but does not fully meet PLOS ONE’s publication criteria as it currently stands. Therefore, we invite you to submit a revised version of the manuscript that addresses the points raised during the review process.

You have received 3 reviews of your paper, each of which is raises some very important points that need to be addressed before this paper could be considered acceptable for publication in PlosOne.  You are therefore invited to revise your paper in the light of the reviewers’ comments.  All comments must be address in the revised paper, with clear indication of the modified text, or in an accompanying letter of rebuttal.

We look forward to receiving your revised manuscript.

Kind regards,

Maura (Gee) Geraldine Chapman, PhD DSc

Academic Editor

PLOS ONE

Journal Requirements:

Additional Editor Comments:

You have received 3 reviews of your paper, each of which is raises some very important points that need to be addressed before this paper could be considered acceptable for publication in PlosOne. You are therefore invited to revise your paper in the light of the reviewers’ comments. All comments must be address in the revised paper, with clear indication of the modified text, or in an accompanying letter of rebuttal.

Reviewers' comments:

Reviewer's Responses to Questions

**Comments to the Author**

1. Is the manuscript technically sound, and do the data support the conclusions?

Reviewer #1: Partly

Reviewer #2: Partly

Reviewer #3: Partly

2. Has the statistical analysis been performed appropriately and rigorously? 

Reviewer #1: Yes

Reviewer #2: I Don't Know

Reviewer #3: Yes

3. Have the authors made all data underlying the findings in their manuscript fully available?

Reviewer #1: Yes

Reviewer #2: Yes

Reviewer #3: Yes

4. Is the manuscript presented in an intelligible fashion and written in standard English?

Reviewer #1: Yes

Reviewer #2: Yes

Reviewer #3: Yes

5. Review Comments to the Author

Reviewer #1: This is an interesting topic although there is a slight disjunct between the majority of the results/discussion and the focus in the introduction. The introduction leads a reader to believe that the majority of the results and discussion will be about niche construction and changes at an evolutionary scale to bacterial composition. However the majority of the results and the discussion are related to the effects of engineering species on diatom community composition, bacterial community composition and functioning and some biogeochemistry of the sediment. I suggest the introduction spends more time on these aspects and sets up some hypotheses- this could have the effect of allowing the (very) long discussion to be shortened. Indeed the ability of the engineers to create selection processes is inferred rather than demonstrated (e.g., L104-107 "A positive response by microbial communities would be observed in the shift in relative abundance of taxa; if the selection pressures exerted by ecosystem engineering is powerful enough to affect community composition, it is implied that they will also influence natural selection at the organism level." and L378 - 380 "Whilst this is not empirical proof of niche construction affecting genomic change, it is expected to occur if community change due to biochemical selection pressure arises."

The results are also more complicated than may have been necessary, consisting of the reporting of a number of ANOSIMS with no interaction terms. If PERMAnova with interactions (e.g., depth*treatment) had been used then the number of tests to be reported may have been less

Reviewer #2: The paper reports an experiment in mesocosm where 2 species of infauna invertebrates were added to the sediment in order to understand their effect on microphytobenthos and bacteria as well as to test for changes in a series of sediment functions. My feeling is that the paper has a good data base but it needs still a bit more thinking and working.

I do not understand how the evolutionary niche concept was analysed. This is a very innovative approach to the study of ecosystem engineers but, besides the nice explanation in the introduction, I do not understand how this is measured or tested in the paper and I doubt that the experiment length was enough to have any idea about how bioturbation can drive evolution of microbial community. However, I am not a microbiologist and I might have got wrong this point

In the aims it would be nice to understand more clearly which are the expectations from the different treatment levels. Why the two species treatment should differ from the 1 species treatment and why and how the two 1-species treatment levels should differ. What would be the expectation for the manual turbation ?

It is also not clear from the paper whether species diversity (the 2-species treatment) was of any interest. In this case I wonder why a two species treatment was added.

The experimental design is simple, but relatively correct, with 4 indipendent mesocosms for each of the 4 treatment levels, 2 with one of the 2 species, 1 with both species , 1 for control and 1 for the control of physical bioturbation. The biomass of the animals was kept constant, so that half biomass of each species was added to the 2 species treatment level. Now, in my opinion, the experiment should also have had a control with reduced biomass of each species to test for diversity effect.

There is missed information concerning the number of individuals that were put in each mesocosm. Only the total biomass is reported. However, the size of individuals is also of great importance for determining the bioturbation effect. It would also be of great interest to have an idea of the mortality rate of individuals during the 2 months of incubation

The « manual turbation » treatment level is a bit unclear. I appreciated this idea, but I am not sure this was done in an appropriate way, since the 2 species used have different bioturbation mechanisms, but both create tubes inside the sediment, while the manual bioturbation repeatedly made “holes” from the sediment surface. I am not surprised that this bioturbation was different from the real animals’ treatments.

In the method, I would als like to see more details concerning the statistical analyses, such as test for heteroscedasticity of residuals, normal distribution and also explanation about the model anova used and which aposteriori test was applied. It is also not clear if and how vertical profiles were analysed because they are repeated measures. In addition, were the differences for univariate indicators of diatoms assemblages analysed statistically or not ?

Small comments:

All the figures with PCO are a bit confusing, I would suggest clearer more detailed captions and also maybe put a title on each PCO

The table 1 order should follow the order in the text.

The order of figures is inverted (we go from figure 5 to figure 1)

Reviewer #3: This experimental study compares the effect of physical and biological disturbance on algal and bacterial community compositions and functions in the sediment from intertidal mudflat.

The topic is interesting and method used relatively novel bringing new and useful knowledge in this field.

I have several concerns that should be addressed before publication of this study. Most of them could be considered in a new version (but I doubt for the first one).

Major concerns

1) My major concern is linked with the experimental design.

As said in the title of the paper, this study deals with “intertidal sediments”.

Reasons for this choice are not clearly presented in the introduction. Intertidal sediments constitute environments with highly fluctuating physical conditions (T°, salinity, oxygen…) at really small times scales. In those highly fluctuating environments, consequences of the addition of biological disturbance should be less visible than in more stable environments. Reasons for this choice should be clearly presented as is does not sound relevant to me.

But the main problem is that those intertidal conditions were not mimicked during the experiment. Information about this aspect are elusive “water changes of 75% overlaying water volume were performed every 3-4 d initially, with longer intervals as the incubation progressed (line 136)”. Those incubation conditions are far from environmental conditions and in such case, conclusions from the study or not realistic with real field processes.

2) In order to differentiate effects from physical and biological activity of bioengineers, sediment was manually perturbated. This manual-turbation (line 137) is described at length (maybe it would be helpful to add an additional figure with a sketch of this apparatus).

This apparatus looks new and was never used before as references about his use are not given. In such case, the reliability of this apparatus is not known.

Does it really mimic physical perturbation of amphipod?

How the frequency of bioturbation (5 complete turns every 24 hours) was chosen?

Does it is realistic with abundance of Corophium used?

Those answers should be given in order to compare Corophium biological perturbation and manual-turbation.

3) One treatment is called “MPB”. I find this word misleading as microphytobenthos was present in other treatments (in lower abbudance). For instance, meiofauna was potentially affected by treatment but was not evaluated. In such context why choosing MPB rather than meiofauna? I would recommend to use “absence of biological and physical turbation”

Minor concerns

Line 56: I have trouble understanding the this sentence

Line 104: why this response will be “positive” and not “negative”… I would only use “response”

Line 114: see major concern #3 about MPB

Line 120: country were samples were collected should be at least be given (°lat and long are not easy to read)

Line 120: biological and biochemical processes are strongly linked with granulometry. In order to give the opportunity to compare future studies with the present one, sediment granulometry should be given

Line 120: tidal elevation is not given. As previously said (major concern #1), this elevation is highly influencing this environment.

Line 131: how does those abundances were chosen? Are they realistic? Those aspects should be discussed later in the paper

Line 134: What was the frequency of death of infauna. If this death rate is high that suggest that incubations conditions are not optimal for those species. In such case, behavior can be affected and results of the present study would not be realistic

Line 137: see major concern #2

Line 418: remove .

Line 513: at world scale?

Line 521: I would use the word “potential” instead of “interesting”

Line 533: mudflat are not characterized by high diversity I would change the formulation

Line 555: some journal names are given with abbreviations whereas others are not

Line 603: “exopolymers”

Line 633: Name of journal, volume and pages are lacking

Line 691: volume and pages are lacking

Fig 5: the resolution does not allow to read results

6. PLOS authors have the option to publish the peer review history of their article (what does this mean?). If published, this will include your full peer review and any attached files.

Reviewer #1: No

Reviewer #2: No

Reviewer #3: No

---

## [Author Response · Author response to Decision Letter 0]

27 Dec 2020

Response to Reviewers

The authors would like to thank the reviewers and editor for their constructive criticism to improve the manuscript. Line numbers refer to final manuscript. 

Comments 

Editor

• In your Methods section, please provide additional information regarding the permits you obtained for the work. Please ensure you have included the full name of the authority that approved the field site access and, if no permits were required, a brief statement explaining why.

o Lines 114-116 added: Permission was obtained to collect samples at the River Tay Site of Special Scientific Interest (SSSI) from Scottish Natural Heritage under Section 16(3) of the Nature Conservation (Scotland) Act 2004.

• Please note location of Rhodes University change from Grahamstown to Makhanda due to official name change.

Reviewer #1: 

• This is an interesting topic although there is a slight disjunct between the majority of the results/discussion and the focus in the introduction. The introduction leads a reader to believe that the majority of the results and discussion will be about niche construction and changes at an evolutionary scale to bacterial composition. However the majority of the results and the discussion are related to the effects of engineering species on diatom community composition, bacterial community composition and functioning and some biogeochemistry of the sediment. I suggest the introduction spends more time on these aspects and sets up some hypotheses- this could have the effect of allowing the (very) long discussion to be shortened. Indeed the ability of the engineers to create selection processes is inferred rather than demonstrated (e.g., L104-107 "A positive response by microbial communities would be observed in the shift in relative abundance of taxa; if the selection pressures exerted by ecosystem engineering is powerful enough to affect community composition, it is implied that they will also influence natural selection at the organism level." and L378 - 380 "Whilst this is not empirical proof of niche construction affecting genomic change, it is expected to occur if community change due to biochemical selection pressure arises."

o Several sections of the introduction that focused a lot on niche construction have been shortened or removed completely. 

o Section on aims and objectives made clearer at the end of the introduction. 

o Section in discussion about the specific niches that were created (L403-410)

• The results are also more complicated than may have been necessary, consisting of the reporting of a number of ANOSIMS with no interaction terms. If PERMAnova with interactions (e.g., depth*treatment) had been used then the number of tests to be reported may have been less

o The format of the analyses was chosen in order to compare surface sediments to each other, and then those at certain depths with each other to compare treatment effects at those depths. The 2-way interaction model was not discussed at length as the comparison between (for example) surface sediments of the MPB treatment and 45 mm sediments of the manual turbation treatment were not of great interest. The 2-way interaction is briefly discussed at lines L325-335.

o ANOSIM results are now in tables in supplementary materials for clarity on analyses. 

Reviewer #2: 

I do not understand how the evolutionary niche concept was analysed. This is a very innovative approach to the study of ecosystem engineers but, besides the nice explanation in the introduction, I do not understand how this is measured or tested in the paper and I doubt that the experiment length was enough to have any idea about how bioturbation can drive evolution of microbial community. However, I am not a microbiologist and I might have got wrong this point

In the aims it would be nice to understand more clearly which are the expectations from the different treatment levels. Why the two species treatment should differ from the 1 species treatment and why and how the two 1-species treatment levels should differ. What would be the expectation for the manual turbation ?

• It is also not clear from the paper whether species diversity (the 2-species treatment) was of any interest. In this case I wonder why a two species treatment was added.

o Lines 101-102 added for justification of the 2 species interaction. The 2 species interaction is also discussed in the discussion. 

o Lines 103-105 indicate expectations from manual turbation, further description inserted at lines 149-151: This process was not designed to precisely mimic the actions of infauna, rather it was to examine the effects of physical disturbance and sediment reworking without macrofaunal burrow irrigation. 

o Clarity on experiment expectations regarding niche concept given throughout.

• The experimental design is simple, but relatively correct, with 4 indipendent mesocosms for each of the 4 treatment levels, 2 with one of the 2 species, 1 with both species , 1 for control and 1 for the control of physical bioturbation. The biomass of the animals was kept constant, so that half biomass of each species was added to the 2 species treatment level. Now, in my opinion, the experiment should also have had a control with reduced biomass of each species to test for diversity effect.

o This is certainly an avenue for future research. This was not included as the focus was differences between infauna types. Density-dependent effects are also hard to decipher as density does not always equal burrowing activity or bioturbation rates. 

• There is missed information concerning the number of individuals that were put in each mesocosm. Only the total biomass is reported. However, the size of individuals is also of great importance for determining the bioturbation effect.

o Numbers of individuals were not counted as this was impractical for the mesocosm setup regarding infauna stress and the large numbers of corophium involved. Previous studies either refer to animal number or biomass, rarely both, and biomass was selected as the metric to use for mesocosm setup. Text added at line 129: Infauna densities were similar to previous mesocosm studies (4), with 1.10 mg infauna per cm3 of sediment, and 10.6 mg infauna per cm2 of sediment surface area. Exact number of individuals were not recorded to minimise handling of infauna, however 3.5 g was equivalent to roughly 750 individuals of C. volutator, and 35 H. diversicolor.

o Information also added about mesocosm aeration. Line 155: Mesocosms were aerated using traditional fish tank air stones, ensuring oxygen saturation at all times. Bubbles were diverted at a 45 ° angle to create a gentle rotational current.

• It would also be of great interest to have an idea of the mortality rate of individuals during the 2 months of incubation

o Information on mortality given in methods: Line 134-138: Daily mortality rates of H. diversicolor were low with no mass mortality events. C. volutator were generally low (<5%), with intermittent higher mortality (first event at 26 days of ~<30%) presumably due to the expected high mortality rates during moulting events observed in C. volutator and other Corophium species in captivity (31, 32).

• The « manual turbation » treatment level is a bit unclear. I appreciated this idea, but I am not sure this was done in an appropriate way, since the 2 species used have different bioturbation mechanisms, but both create tubes inside the sediment, while the manual bioturbation repeatedly made “holes” from the sediment surface. I am not surprised that this bioturbation was different from the real animals’ treatments.

o Intention of treatment clarified: Line 149-151: This process was not designed to precisely mimic the actions of infauna, rather it was to examine the effects of physical disturbance and sediment reworking without macrofaunal burrow irrigation. 

o Treatment effectiveness discussed: Line 420-429: The manual turbation of sediments using the previously described apparatus (Supp. Fig. 1) was successful in recreating the sediment reworking aspects of bioturbation without the bioirrigation of sediments. Turbidity was comparable to the C. volutator treatment, with very similar values for the majority of the experiment. In the final stages of the experiment turbidity in the manually turbated mesocosms were higher, presumably due to the decrease in sediment reworking by C. volutator as burrows had become established and were stable. Dissolved oxygen concentrations in the manually turbated sediments below the surface were negligible, whereas treatments containing infauna had higher concentrations due to burrow irrigation. The redox potential of sediments were similar between the manually turbated sediment and those containing infauna, demonstrating that redox was driven primarily by sediment reworking. 

• In the method, I would als like to see more details concerning the statistical analyses, such as test for heteroscedasticity of residuals, normal distribution and also explanation about the model anova used and which aposteriori test was applied.

o ANOVA model information removed from table caption and now in table. 

o Will check residuals etc again and put in sentence saying they were checked. Also sentence about fishers LSD for anovas, and Dunns for kruskall wallis test

• It is also not clear if and how vertical profiles were analysed because they are repeated measures. 

o No formal analyses were performed on the vertical profiles. 

• In addition, were the differences for univariate indicators of diatoms assemblages analysed statistically or not ?

o Please note corrections to Evenness values. 

o Kruskall wallis test added to table 2. 

o Text added to line 262: There were no significant differences between the alpha diversity metrics of the diatom communtities, with the exception of C. volutator and manual turbation mesocosms displaying a higher Pielou’s evenness of species than H. diversicolor and MPB mesocosms (Dunn’s post-hoc tests: z = 2.27, p = 0.012; z = 2.21, p = 0.014 for C. volutator vs H. diversicolor and MPB respectively. z = 2.31, p = 0.010; z = 2.7, p = 0.012 for manual turbation vs H. diversicolor and MPB respectively).

o The authors thank the reviewer for highlighting the lack of univariate analyses on the communities. 

Univariate statistics have been performed, and results sections altered to reflect these. 

• All the figures with PCO are a bit confusing, I would suggest clearer more detailed captions and also maybe put a title on each PCO

o Titles on PCOs, and captions improved

• The table 1 order should follow the order in the text.

o Table 1 rows rearranged

• The order of figures is inverted (we go from figure 5 to figure 1

o This will be rectified on the new upload. 

• Reviewer #3: 1) My major concern is linked with the experimental design.

As said in the title of the paper, this study deals with “intertidal sediments”.

Reasons for this choice are not clearly presented in the introduction. Intertidal sediments constitute environments with highly fluctuating physical conditions (T°, salinity, oxygen…) at really small times scales. In those highly fluctuating environments, consequences of the addition of biological disturbance should be less visible than in more stable environments. Reasons for this choice should be clearly presented as is does not sound relevant to me.

But the main problem is that those intertidal conditions were not mimicked during the experiment. Information about this aspect are elusive “water changes of 75% overlaying water volume were performed every 3-4 d initially, with longer intervals as the incubation progressed (line 136)”. Those incubation conditions are far from environmental conditions and in such case, conclusions from the study or not realistic with real field processes.

o Sentence added in M&M: Lines 157-160: Although these conditions do not specifically mimic intertidal conditions that the sediments were extracted from, this setup allows the comparison of the niche constructing activities of organisms capable of living inter-and subtidally, with all treatments exposed to the same conditions of good light availability and moderate salinity, similar to previous studies (33).

• 2) In order to differentiate effects from physical and biological activity of bioengineers, sediment was manually perturbated. This manual-turbation (line 137) is described at length (maybe it would be helpful to add an additional figure with a sketch of this apparatus).

o Photo in supplementary materials (Supp. Fig 1)

• This apparatus looks new and was never used before as references about his use are not given. In such case, the reliability of this apparatus is not known. 

o Text inserted, lines 149-151: This process was not designed to precisely mimic the actions of infauna, rather it was to examine the effects of physical disturbance and sediment reworking without macrofaunal burrow irrigation

o Text added to discuss efficacy of manual turbation Lines 42-429: The manual turbation of sediments using the previously described apparatus (Supp. Fig. 1) was successful in recreating the sediment reworking aspects of bioturbation without the bioirrigation of sediments. Turbidity was comparable to the C. volutator treatment, with very similar values for the majority of the experiment. In the final stages of the experiment turbidity in the manually turbated mesocosms were higher, presumably due to the decrease in sediment reworking by C. volutator as burrows had become established and were stable. Dissolved oxygen concentrations in the manually turbated sediments below the surface were negligible, whereas treatments containing infauna had higher concentrations due to burrow irrigation. The redox potential of sediments were similar between the manually turbated sediment and those containing infauna, demonstrating that redox was driven primarily by sediment reworking. 

• Does it really mimic physical perturbation of amphipod? 

o Clarification of manual turbation treatment as mentioned above

• How the frequency of bioturbation (5 complete turns every 24 hours) was chosen?

o This was chosen after preliminary testing showed sufficient reworking was achieved, and a similar overlying water turbidity to that of C. volutator was achieved. The satisfactory similarity to aspects of the infaunal treatments is now discussed in lines 420-429

• Does it is realistic with abundance of Corophium used?

o See text inserted, lines 129-133 and the comment above.

• Those answers should be given in order to compare Corophium biological perturbation and manual-turbation. 

o See above. 

• Line 137: see major concern #2

o Figure added to supplementary material

• 3) One treatment is called “MPB”. I find this word misleading as microphytobenthos was present in other treatments (in lower abbudance). For instance, meiofauna was potentially affected by treatment but was not evaluated. In such context why choosing MPB rather than meiofauna? I would recommend to use “absence of biological and physical turbation” (also see comment below)

• Line 114: see major concern #3 about MPB

o Treatment conditions clarified on line 125-127: Treatment 4 had no infauna added, and no physical turbation to allow MPB biofilms to develop ungrazed by infauna as with treatments 1-3, and undisturbed as with treatment 5, and is hereon referred to as the ‘MPB’ treatment

• Line 56: I have trouble understanding this sentence 

o Sentence changed to: Intertidal flats are spatially homogenous compared to other marine habitats and many terrestrial habitats, however the inhabitant organisms occupy different niches with regards to sediment depth, substratum type or tidal elevation (2).

• Line 104: why this response will be “positive” and not “negative”… I would only use “response”

o “positive” removed.

• Line 120: country were samples were collected should be at least be given (°lat and long are not easy to read)

o “Scotland, UK” added.

• Line 120: biological and biochemical processes are strongly linked with granulometry. In order to give the opportunity to compare future studies with the present one, sediment granulometry should be given

o Added to microcosm setup L111-114: “The sediment collected was a mud and sand mix typical of sediments towards the mouth of estuaries (Particle size distribution: mean= 178 µm; median= 182; % <63= 16%; water content= 24%; organic content= 2%).”

• Line 120: tidal elevation is not given. As previously said (major concern #1), this elevation is highly influencing this environment.

o ‘mid’-intertidal instead of intertidal.

o Not replicating the system where sediments were collected.

• Line 131: how does those abundances were chosen? Are they realistic? Those aspects should be discussed later in the paper

o Text inserted, Line 129-133: Infauna densities were similar to previous mesocosm studies (4), with 1.10 mg infauna per cm3 of sediment, and 10.6 mg infauna per cm2 of sediment surface area. Exact number of individuals were not recorded to minimise handling of infauna, however 3.5 g was equivalent to roughly 750 individuals of C. volutator, and 35 H. diversicolor, equivalent to 24,000 individuals/m2 and 1110 individuals/m2 respectively, which are realistic environmental abundances (23, 30).

• Line 134: What was the frequency of death of infauna. If this death rate is high that suggest that incubations conditions are not optimal for those species. In such case, behavior can be affected and results of the present study would not be realistic

o See comment above and refer to lines

• Line 418: remove .

o Line 459 corrected 

• Line 513: at world scale?

o “globally (59)” added. Reference changed to :Liou JS, Madsen EL. 2008. Microbial ecological processes: Aerobic/anaerobic. In: Jørgensen SE, Fath BD, eds. Encyclopedia of ecology. Oxford: Academic Press; 2348-2357.

• Line 521: I would use the word “potential” instead of “interesting”

o Changed to “potential”

• Line 533: mudflat are not characterized by high diversity I would change the formulation

o Sentence changed to: L572: This reveals the vulnerability of microbial metabolism, as a proxy for key ecosystem functions and services, to changes in macrofaunal assemblages and demonstrates the importance of ecosystem engineers to sustaining functional coastal systems.

• Line 555: some journal names are given with abbreviations whereas others are not

o This has been corrected

• Line 603: “exopolymers”

o Corrected

• Line 633: Name of journal, volume and pages are lacking

o Full reference entered

• Line 691: volume and pages are lacking

o Will check over and rectify all references

• Fig 5: the resolution does not allow to read results

o This should be rectified on re-upload

---

## [Decision Letter · Decision Letter 1]

19 Jan 2021

PONE-D-20-31006R1

Ecosystem engineers drive differing microbial community composition in intertidal estuarine sediment

PLOS ONE

Dear Dr. Wyness,

Thank you for submitting your manuscript to PLOS ONE. After careful consideration, we feel that it has merit but does not fully meet PLOS ONE’s publication criteria as it currently stands. Therefore, we invite you to submit a revised version of the manuscript that addresses the points raised during the review process.

Academic Editor

I will be happy to accept this manuscript for publication if you can consider addressing the one remaining  minor comment raised by Reviewer 2.

We look forward to receiving your revised manuscript.

Kind regards,

Maura (Gee) Geraldine Chapman, PhD DSc

Academic Editor

PLOS ONE

Additional Editor Comments (if provided):

Academic Editor

I will be happy to accept this manuscript for publication if you can consider addressing the one remaining minor comment raised by Reviewer 2.

Reviewers' comments:

Reviewer's Responses to Questions

**Comments to the Author**

1. If the authors have adequately addressed your comments raised in a previous round of review and you feel that this manuscript is now acceptable for publication, you may indicate that here to bypass the “Comments to the Author” section, enter your conflict of interest statement in the “Confidential to Editor” section, and submit your "Accept" recommendation.

Reviewer #2: All comments have been addressed

Reviewer #3: All comments have been addressed

2. Is the manuscript technically sound, and do the data support the conclusions?

Reviewer #2: Yes

Reviewer #3: Yes

3. Has the statistical analysis been performed appropriately and rigorously? 

Reviewer #2: Yes

Reviewer #3: I Don't Know

4. Have the authors made all data underlying the findings in their manuscript fully available?

Reviewer #2: No

Reviewer #3: Yes

5. Is the manuscript presented in an intelligible fashion and written in standard English?

Reviewer #2: Yes

Reviewer #3: Yes

6. Review Comments to the Author

Reviewer #2: the paper has improved and comments have been addressed. I appreciate that niche concept has been diminished of importance. In my opinion I think that sticking witht the concept of ecosystem engineer is enough and clear. I have only one comment that concerns the aims and the experiment. Indeed, microphytobenthos is considered an explanatory variable and also a response variable. Could this point be adressed a bit better?

Reviewer #3: (No Response)

7. PLOS authors have the option to publish the peer review history of their article (what does this mean?). If published, this will include your full peer review and any attached files.

Reviewer #2: No

Reviewer #3: No

---

## [Author Response · Author response to Decision Letter 1]

20 Jan 2021

Response to reviewers (2)

Reviewer #2: the paper has improved and comments have been addressed. I appreciate that niche concept has been diminished of importance. In my opinion I think that sticking witht the concept of ecosystem engineer is enough and clear. I have only one comment that concerns the aims and the experiment. Indeed, microphytobenthos is considered an explanatory variable and also a response variable. Could this point be adressed a bit better?

This is a well-raised issue, that it is currently framed as both an explanatory and response variable. 

In order to clarify this, the sentence:

Line 93: ‘Finally, by not adding macrofauna to mesocosms, the effect of microphytobenthos (MPB), were also studied.’

Has been changed to: 

Finally, by not adding macrofauna to mesocosms, the effect of an uninterrupted microphytobenthos (MPB) biofilm, was also studied.

This highlights the fact that the MPB biofilms are present in the other treatments, and it is the lack of macrofauna that is the treatment, and the study of the MPB biofilm as a response variable is valid. 

Also, at line 126: ‘Treatment 4 had no infauna added, and no physical turbation to allow MPB biofilms to develop ungrazed by infauna as with treatments 1-3, and undisturbed as with treatment 5, and is hereon referred to as the ‘MPB’ treatment.’ further highlights this point, and notes explicitly ‘MPB’ as an explanatory variable is again an absence of infauna, and is labelled as MPB for ease of understanding.

---

## [Editor Report · Decision Letter 2]

25 Jan 2021

Ecosystem engineers drive differing microbial community composition in intertidal estuarine sediments

PONE-D-20-31006R2

Dear Dr. Wyness,

We’re pleased to inform you that your manuscript has been judged scientifically suitable for publication and will be formally accepted for publication once it meets all outstanding technical requirements.

Kind regards,

Maura (Gee) Geraldine Chapman, PhD DSc

Academic Editor

PLOS ONE
---

## [Editor Report · Acceptance letter]

1 Feb 2021

PONE-D-20-31006R2 

Ecosystem engineers drive differing microbial community composition in intertidal estuarine sediments. 

Dear Dr. Wyness:

I'm pleased to inform you that your manuscript has been deemed suitable for publication in PLOS ONE. Congratulations! Your manuscript is now with our production department. 

Kind regards, 

on behalf of

Professor Maura (Gee) Geraldine Chapman 

Academic Editor

PLOS ONE